# Moisture-triggered fast crystallization enables efficient and stable perovskite solar cells

Kaikai Liu[1,3], Yujie Luo[1,3], Yongbin Jin[1], Tianxiao Liu[2], Yuming Liang[1], Liu Yang[1], Peiquan Song[1], Zhiyong Liu[2], Chengbo Tian[1], Liqiang Xie ®[1] ✉ & Zhanhua Wei ®[1] ✉

Understanding the function of moisture on perovskite is challenging since the random environmental moisture strongly disturbs the perovskite structure. Here, we develop various $N_2$-protected characterization techniques to comprehensively study the effect of moisture on the efficient cesium, methylammonium, and formamidinium triple-cation perovskite $(Cs_{0.05}FA_{0.75}MA_{0.20})Pb(I_{0.96}Br_{0.04})_3$. In contrast to the secondary measurements, the established air-exposure-free techniques allow us directly monitor the influence of moisture during perovskite crystallization. We find a controllable moisture treatment for the intermediate perovskite can promote the mass transportation of organic salts, and help them enter the buried bottom of the films. This process accelerates the quasi-solid-solid reaction between organic salts and $PbI_2$, enables a spatially homogeneous intermediate phase, and translates to high-quality perovskites with much-suppressed defects. Consequently, we obtain a champion device efficiency of approaching 24% with negligible hysteresis. The devices exhibit an average $T_{80}$-lifetime of 852 h (maximum 1210 h) working at the maximum power point.

Depositing high-quality perovskite films with few defects plays a crucial role in achieving efficient and stable perovskite solar cells (PSCs)[1–7]. Perovskite bulk doping and interface modification have been demonstrated to be effective in suppressing the bulk and interfacial defects[8–13]. It is reported that the perovskite formation process is sensitive to the surrounding atmosphere, like the solvent vapor, moisture, and others[14–18]. For example, for organic-inorganic halide perovskite materials containing methylammonium (MA), many previous reports claimed that moisture could penetrate the bulk of perovskite film, destroy the organic-inorganic interactions, and generate the undesired $PbI_2$, resulting in the loss of photovoltaic properties[19]. The disruptive effect of $H_2O$ on perovskite was ascribed to the irreversible hydration process, leading to the degradation of perovskite to

$PbI_2$. However, when the dose of $H_2O$ was low, this hydration process was reversible, and the hydrated product could spontaneously dehydrate in a dry atmosphere[20]. This interesting phenomenon inspired researchers to explore the dual role of $H_2O$ in perovskite materials and the resulting devices.

The impact of $H_2O$ on perovskite materials may occur everywhere during the device fabrication, including weighing raw materials, preparing precursor solution, spin-coating, and annealing[21–29]. For instance, $H_2O$ molecules could tune the morphology of the $PbI_2$ film. It was observed that exposing the mesoporous $TiO_2$ films to moisture for three minutes before spin-coating could lead to the formation of a porous $PbI_2$ structure, enhancing the subsequent reaction between MAI and $PbI_2$[21]. It was also deemed that $H_2O$

[1]Xiamen Key Laboratory of Optoelectronic Materials and Advanced Manufacturing, Institute of Luminescent Materials and Information Displays, College of Materials Science and Engineering, Huaqiao University, Xiamen 361021, P.R. China. [2]Henan Key Laboratory of Photovoltaic Materials, School of Physics, Henan Normal University, Xinxiang 453007, P.R. China. [3]These authors contributed equally: Kaikai Liu, Yujie Luo. ✉e-mail: lqxie@hqu.edu.cn; weizhanhua@hqu.edu.cn

molecules could directly take part in the chemical reaction to form perovskite. $H_2O$ molecules can be reversibly absorbed on the surface of the as-prepared perovskite film, facilitating the conversion of the unreacted ions, leading to high-quality perovskite films[23]. Besides, Adhikari et al. claimed that tiny amounts of $H_2O$ incorporated into the MAI solution could accelerate the crystal growth of perovskite film through forming the metastable $CH_3NH_3PbI_3 \cdot H_2O$ or $(CH_3NH_3)_4PbI_6 \cdot 2H_2O$, which was favorable to grow perovskite films with fewer defects[25]. Moreover, it was reported that $H_2O$ molecules could repair the imperfections in perovskite film via the dissolution and recrystallization of perovskite grains when annealing perovskite film in the air[26]. Huang et al. utilized in-situ grazing-incident wide-angle X-ray scattering (GIWAXS) to capture the phase transformation of the $CH_3NH_3I_xCl_{3-x}$ perovskite films during annealing under different relative humidity. They concluded that a moderate water content accelerates the crystal formation and enhances the texture-orientation of the film[29]. These reports imply that $H_2O$ molecules may regulate the film structure or the crystal growth, thus affecting the final film quality.

It is noted that many previous reports have investigated the role of $H_2O$ in improving the quality of perovskite films (summarized in Supplementary Table 1). However, the investigated effect of $H_2O$ is always coupled with other factors such as the surrounding atmosphere, the heating thermal energy, and the solvent, which depends on the processing strategies. Moreover, most of the conclusions were even drawn from studying the final device performance, which is indirect evidence influenced by many complicated factors. Decoupling the actual role of moisture from these factors is essential to obtain the so-called 'direct evidence' of the water-induced effects. Systematic studies monitoring the intermediate perovskite treated by moisture alone are more convincing. Notably, the freshly prepared intermediate perovskite films before annealing ("wet film") are metastable and sensitive to the surrounding environment. Therefore, investigating the effect of $H_2O$ on them strictly requires more careful experimental designs. Moreover, the effect of water depends on the perovskite composition[30]. As the cesium, MA, and formamidinium (FA) triple-cation perovskite possessing a low bromide/iodide ratio delivers higher theoretical efficiency, gaining insights into the effect of moisture on this system may contribute to a more purposeful design of high-performance PSCs that simultaneously possess high efficiency and robust stability.

Here, by protecting and characterizing the metastable perovskite samples with various $N_2$-filled cabins, we managed to investigate the actual effect of $H_2O$ (moisture in the air) on the intermediate and final perovskite films with the composition of $(Cs_{0.05}FA_{0.75}MA_{0.20})$ $Pb(I_{0.96}Br_{0.04})_3$ (Supplementary Note 1). Specifically, we monitored the evolution of these wet films before and after moisture treatment by UV-vis spectroscopy (UV-vis), steady-state photoluminescence (PL), X-ray photoelectron spectroscopy (XPS), X-ray diffraction (XRD), scanning electron microscopy (SEM), nuclear magnetic resonance (NMR), thermogravimetric analysis (TGA), etc. We found that the humidified film can trigger the quasi-solid-solid reaction between organic salts and $PbI_2$, realizing a rapid homogeneous distribution of the $PbI_2$-organic salts complex. Consequently, the resultant perovskite films exhibited a more homogenous perovskite phase with larger crystal grains and higher crystallinity. Moreover, the deep-level defects in the resultant perovskite films were much suppressed. As a result, we obtained a champion PCE of 23.93% with prolonged operational stability, remaining 80% of its initial PCE after continuous illumination under 1 sun illumination over 1200 h.

## Results

### Developing air-exposure-free characterization techniques

Figure 1 illustrates three examples of our designed air-exposure-free characterization techniques. The samples to be characterized are the as-prepared wet intermediate perovskite film w and w/o moisture treatment. We adopted inert-gas protected chambers in various configurations to ensure $N_2$-protected measurements (optical spectroscopy, SEM, and XRD). These techniques are essential for the characterization of the wet intermediate perovskite film, which is sensitive to the ambient atmosphere. Specifically, we assembled the optics for spectroscopy measurements and adjusted the optical path in an $N_2$-filled glove box. So inert-gas protected UV-vis absorption and PL spectra (Fig. 1a) could be recorded as soon as the films were prepared in the same glove box. This home-assembled setup (Supplementary Fig. 1) allows us to study the optical properties of the metastable wet intermediate perovskite film. For the morphological measurement, we used a customized shuttle cabin for SEM (Fig. 1b and

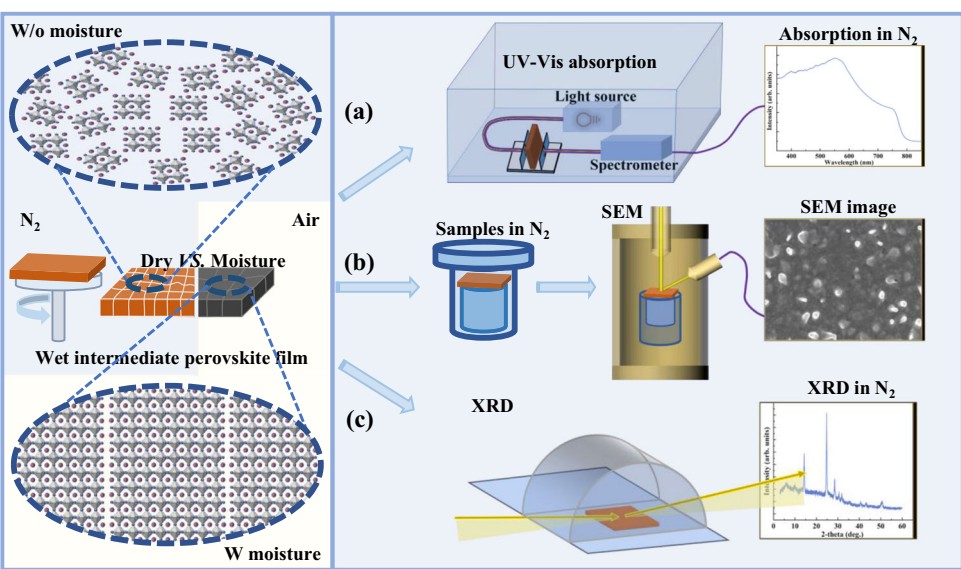

**Fig. 1 | Schematic diagram of the representative N2-protected characterization techniques. a** UV-vis absorption, **b** SEM, and **c** XRD. All the sample preparation and sealing are carried out in the $N_2$-filled glovebox. The UV-vis absorption measurement is carried out in the $N_2$-filled glovebox. The as-prepared intermediate perovskite films are sealed in a transfer chamber and loaded into the SEM directly for characterization. The as-prepared intermediate perovskite films are placed in an $N_2$-filled protective cap for the XRD measurement.

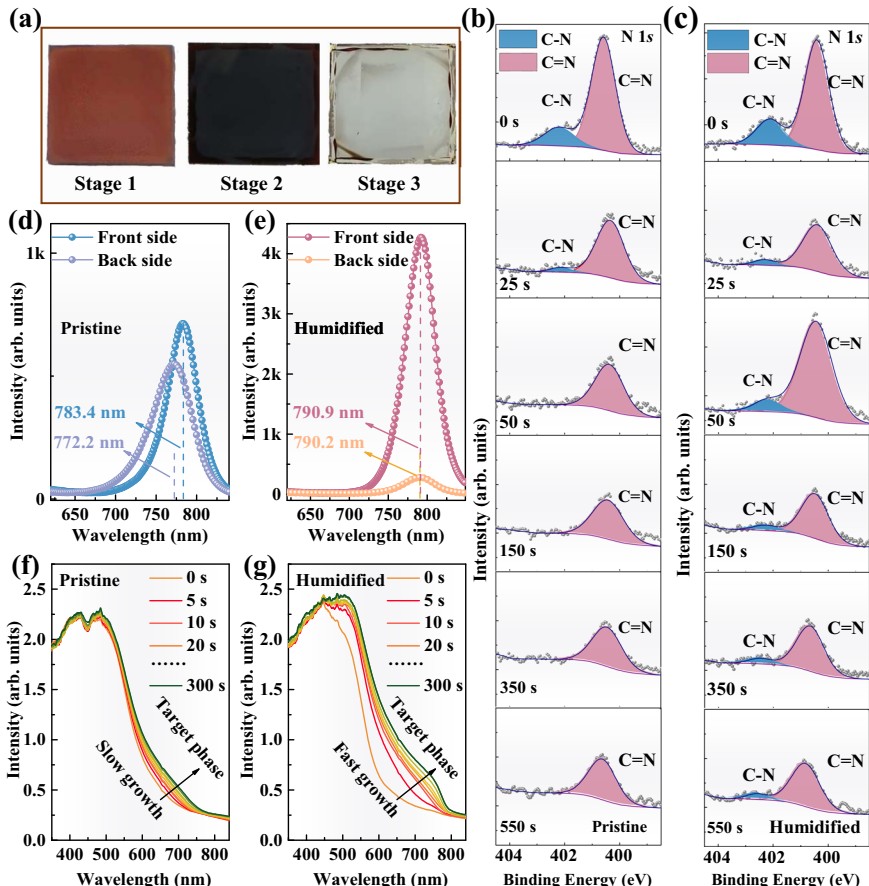

**Fig. 2 | Photoelectric characterization of the wet intermediate perovskite films (before annealing) without and with the moisture treatment. a** Photographs of wet intermediate perovskite film for pristine, humidified, and over humidified samples. **b, c** Depth profile of the XPS spectra of N 1s peak on the pristine and humidified samples. Ar⁺ ion beam is used to etch film samples for desired sputter duration to evaluate the composition distribution of the intermediate film in different depths. **d, e** Steady-state photoluminescence spectra excited from the quartz substrate side (backside) and the perovskite film side (front side). **f, g** Evolution of UV-vis absorption spectra as a function of time for the pristine and the humidified wet films.

Supplementary Fig. 2). This setup enables us to transfer the as-prepared intermediate perovskite film to the SEM directly without any exposure to the ambient air, realizing the observation of the actual morphology for samples w/o and with the moisture treatment. An acrylic semicircular chamber (Fig. 1c) was used to protect the samples during XRD measurements, and all samples were encapsulated with the sealing tape (Supplementary Fig. 3). It can provide an $N_2$ atmosphere for the wet intermediate perovskite film during measurements, avoiding the invasion of moisture in the air. Benefiting from these protective characterization techniques (other protective characterization techniques will be described when used in the following text), we expect to investigate the actual influence of moisture on perovskite.

### The role of moisture treatment in accelerating the crystallization of the as-prepared intermediate perovskite films before annealing

The optical images of three wet intermediate perovskite films prepared via different processing conditions are shown in Fig. 2a. We examined the as-prepared film without moisture treatment (Stage 1, Pristine), intermediate perovskite film with moisture treatment in the air with controlled humidity of 35% (Stage 2, Humidified), and intermediate perovskite films with excess moisture treatment (Stage 3, Over humidified). After spin-coating the organic salt on the $PbI_2$ film (yellow), the films showed an orange-red color. Interestingly, the films immediately turned into shiny-black color when exposed to the in-lab air with 35% relative humidity, and finally changed to dark-black color

within 1 min. These results suggest that $H_2O$ molecules may take part in the reaction between organic salt and $PbI_2$ in the intermediate film and could accelerate the crystallization. This phenomenon indicates that more perovskite crystallization sites formed in the humidified film, which is beneficial for forming a high-quality perovskite film[31]. The over-humidified perovskite films (moisture treatment with more than 3 min) displayed a transparent appearance, possibly due to the complete coordination of water molecules to perovskite[20]. Besides, when the same wet intermediate perovskite film was exposed to the air for different stages, similar color changes were observed (Supplementary Fig. 4). Primary solar cell screening experiments revealed that the over-humidified films resulted in inferior device performances, so we will mainly focus on the pristine and humidified samples hereafter.

To investigate the effect of moisture on the spatial distribution of the chemical species in the intermediate perovskite film, we performed depth-dependent XPS measurements on the wet intermediate perovskite films without (Pristine) and with (Humidified) moisture treatment (Fig. 2b, c). The C = N bond (400.07 eV) and C-N (402.14 eV) bond are assigned to FA and MA, respectively, which can be confirmed by the N 1s spectra of FAI and MAI (Supplementary Fig. 5). As shown in Fig. 2b, for the pristine film, although the FA signal can be detected across the whole thickness (sputtering for 550 s), the MA signal can only be detected on the surface (in the first 25 s). In contrast, for the humidified film (Fig. 2e), the MA signal can be synchronously detected with the FA signal across the whole thickness, indicating MA cation can penetrate deeper into the intermediate perovskite film after moisture treatment. These results imply that moisture can facilitate the

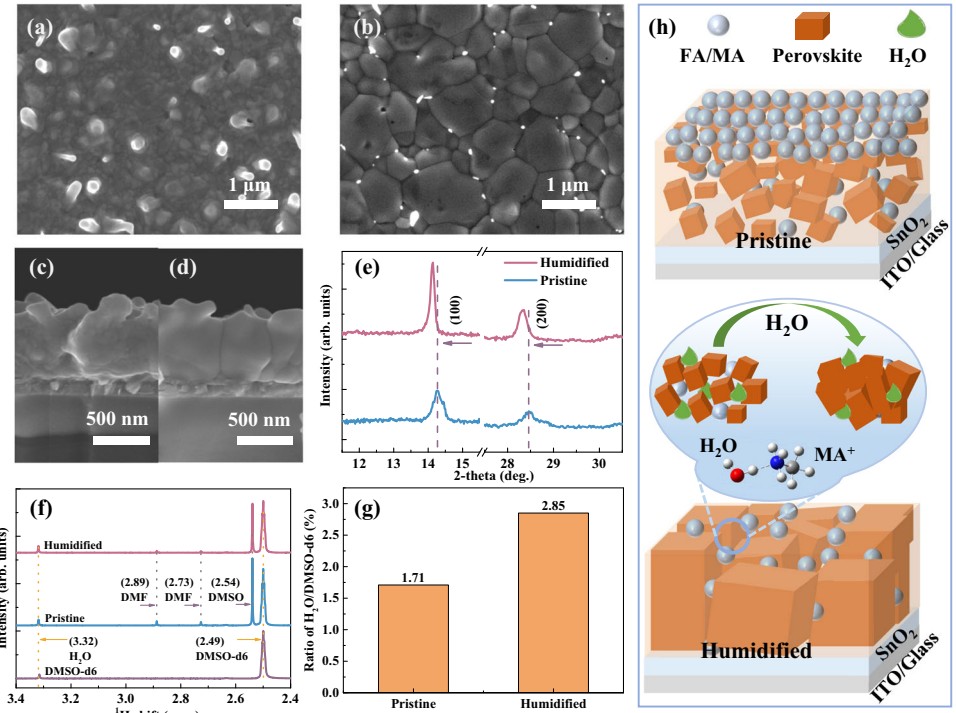

**Fig. 3 | The effect of moisture treatment on the morphology and structure of the wet intermediate perovskite film before annealing.** The morphology evolution and crystallization change of the same intermediate perovskite film from pristine state to humidified state: **a**, **b** Top-view SEM images; **c**, **d** Cross-sectional SEM images; **e** XRD patterns. **f** $^1$H NMR spectra of DMSO-d6, the pristine intermediate perovskite film, and the humidified intermediate perovskite film. **g** The ratio of the relative integral area of $H_2O$ molecules and DMSO-d6 molecules from NMR results. **h** Schematic illustration of $H_2O$ in regulating the crystallization process.

diffusion of organic cations through the intermediate perovskite film in the vertical direction (Supplementary Fig. 6).

The spatial distribution of the chemical species determines the quality of the perovskite films. To examine the transformation process of $PbI_2$ to perovskite, we recorded the steady-state PL spectra of the pristine and humidified intermediate perovskite films (deposited on quartz substrates). The PL signals were acquired by exciting from both the front and the back sides (Supplementary Fig. 7). In Fig. 2d, the pristine sample displays a PL peak located at 772 nm excited from the back side, which is significantly different from the front side (~783 nm), indicating the inhomogeneous distribution of perovskite phases in the perpendicular direction[32]. In contrast, the PL spectra of the humidified sample excited from the front side and the back side show a coincident peak position at about 791 nm (Fig. 2e), confirming the more homogenous distribution of perovskite phases. As for the significant difference in PL intensities, it can be ascribed to the different crystallinity of perovskite film in the upper and bottom parts. Moreover, compared to the PL peak located at ~783 nm for the pristine sample, the humidified sample demonstrates a significant redshift with a PL peak at ~791 nm, suggesting the compositional difference in perovskite phases. These results can be ascribed to the fact that the bromide-related species are preferential to crystalize than iodide-related species during perovskite formation[33], and moisture treatment will promote the subsequent iodide species crystallization.

To further assess the role of moisture treatment on the crystallization kinetics of the perovskite phase, in situ UV-vis absorption spectra over time for the pristine and humidified intermediate perovskite films w/o (Pristine) and with (Humidified) moisture treatment was recorded. In Fig. 2f, the pristine film displays no evident absorption changes in the region between 750 and 800 nm as a function of time. However, the absorption of the humidified film exhibits a distinct difference between 750 and 800 nm, with a strong redshift of band edges in a short period of 60 s (Fig. 2g). We compared the $\Delta A$ (the

increase of absorbance at 780 nm with time relative to the initial value) and $k$ ($\Delta A/t$) to reveal the perovskite formation process[34]. Compared with the pristine film, the humidified film shows a faster absorbance increase rate with a higher $k$ ($9.0 \times 10^{-4}\,s^{-1}$ for humidified film vs. $2.0 \times 10^{-4}\,s^{-1}$ for the pristine film) during the growth process, which confirms the role of moisture treatment in accelerating the formation of perovskite crystallization sites (Supplementary Fig. 8).

In the sequential two-step deposition process to fabricate perovskite films, the as-deposited film usually shows a stacked double-layer structure, and further thermal annealing is required to obtain the uniform perovskite films[35]. Due to the spatial separation of $PbI_2$ and organic salts in the pristine film, the perovskite crystal sites can only form at the interface between $PbI_2$ and organic salts, showing light color with few perovskite crystals. Encouragingly, once the as-deposited film is exposed to moisture, water molecules may form hydrogen bonding with organic salts and improve the mass mobility of hydrated salts[23], which helps the penetration of organic salts into the depth of $PbI_2$ film, leading to the formation of more perovskite crystals. These results further confirm the effect of moisture on the optical properties of perovskites.

## The influence of moisture treatment on morphology and crystal structure of the intermediate perovskite and the final perovskite films

To investigate the effect of moisture on the perovskite crystal formation, we analyzed the evolution of the morphology of the same perovskite film from its initial pristine state to the moisture-treated intermediate perovskite film in different stages (Fig. 3a, b, Supplementary Fig. 9a–d). The pristine film (Fig. 3a) displays a messy grain distribution with many small crystals and protuberances, which can be assigned to a mixture of perovskite and organic salts, respectively. In contrast, when exposed to moisture for 60 s, this humidified state (Fig. 3b) exhibits a homogenous distribution of larger crystal grains.

When further exposing the humidified film to moisture over three minutes, the as-obtained over humidified state exhibits even larger grains (Supplementary Fig. 9a, c), but the homogeneity becomes poor. Although the over-humidified state with the transparent adducts can also reach a similar dark black film after annealing (similar to the annealed pristine one), massive randomly distributed $PbI_2$ on its surface (Supplementary Fig. 9b, d) is unfavorable to the photoelectric properties. We investigated the cross-sectional morphology of the same wet intermediate perovskite films (Fig. 3c, d). It shows a disordered distribution of many small crystal grains in the pristine state. In contrast, the humidified state exhibits much larger crystal grains with the vertically monolithic arrangement, showing the role of moisture treatment in inducing the crystallization process to achieve a better film morphology.

We further investigated the crystal structure and chemical composition of the moisture-treated intermediate perovskite using XRD. The same sample treated with moisture for different times was used to represent different stages. In Fig. 3e, the pristine state exhibits a peak split at 14.3° and 14.5°, corresponding to I-rich and Br-rich perovskite phases, respectively[36]. The peak intensities are lower than the humidified sample, possibly due to fewer perovskite crystal sites and smaller perovskite crystal grains. After moisture treatment, the humidified state displays a narrow and symmetrical peak at 14.1°, indicating that the moisture helps form more perovskite phase and simultaneously eliminates phase segregation. The XRD peaks of the over-humidified state show an obvious shift toward the lower diffraction angle (Supplementary Fig. 10), which can be attributed to the hydrated perovskite phase with saturated water[20].

It is noted that the reproducibility of the optical properties, film morphology and crystal structure of the films is confirmed by double-checked experiments, implying good repeatability of the effect of moisture treatment by the controlled humidity.

The relative content of absorbed water molecules in the different intermediate perovskite films (Pristine and Humidified) was evaluated by the $^1$H-NMR (Fig. 3f). Specifically, we dissolved the as-prepared intermediate perovskite film with DMSO-d6 (solvent in NMR measurements) and measured the corresponding NMR data. The ratio of the integrated area of the characteristic peak between $H_2O$ and DMSO-d6 is used to indicate the relative absorbed water molecules amount. The relative amount of absorbed water molecules is 1.17% and 2.85% for pristine film and humidified film, respectively (Fig. 3g and Supplementary Table 2). The humidified film shows the highest water molecules amount, suggesting the in-lab moisture is indeed introduced into perovskite film after moisture treatment, leading to the changes as mentioned above in the morphology and crystal structure.

Therefore, we summarized the effect of moisture on mass transportation and perovskite crystal formation in Fig. 3h. The incoming water molecules will first interact with the top organic salts (FAI/MAI/MABr/MACl) layer via hydrogen bonds. Then more mobile hydrated salts quickly enter the underneath $PbI_2$ layer, making the perovskite crystallization sites distribute over the entire film other than only at the $PbI_2$-organic salts interface. Moreover, the enhanced mass transportation may also ensure a faster precursor supply required for perovskite growth, resulting in larger perovskite grains (illustrated on the left of Fig. 1)[28].

### Enhancing the quality of perovskite films and the carrier transport in the PSCs via moisture treatment

The top-view SEM image in Fig. 4a shows that the surface of the control perovskite film (the annealed pristine film) is covered by many $PbI_2$ plates (white species in the SEM image). For the target perovskite film (the annealed humidified film), large perovskite grains are tightly packed one by one, and the amount of $PbI_2$ is much reduced (Fig. 4b). AFM results (Supplementary Fig. 11) show that apart from the enlarged perovskite grains, the roughness of the target film (27.6 nm) is lower

than that of the control (30.4 nm). In addition, the cross-sectional SEM images (Supplementary Fig. 12) demonstrate that the control film exhibits an irregular arrangement of small grains while the target film possesses monolithic vertically packed large perovskite grains. The monolithic perpendicular arranged grains and a smooth surface are beneficial for charge transfer in the resultant PSCs[37]. The absorption spectra of the control and the target film (Supplementary Fig. 13) show no obvious difference from each other, indicating no band edge shift after moisture treatment. XRD patterns show the evident characteristic peaks of the alfa-perovskite phase assigned to the crystal planes of (100) and (200) located at 14.2° and 28.4°, respectively (Supplementary Fig. 14). Compared with the control film, the XRD peak of the alfa-perovskite phase in the target film exhibits a threefold enhancement, and that of $PbI_2$ shows a fourfold decrease, suggesting much-enhanced perovskite crystallinity and suppressed $PbI_2$ content in the target film.

We employed various characterizations to evaluate the defect state within the perovskite films, including space-charge limited-current (SCLC)[38], time-resolved photoluminescence (TRPL), and thermal admittance spectroscopy (TAS). As shown in Fig. 4c, the control sample displays a trap-filled limit voltage ($V_{TFL}$) of 1.07 V while the target sample displays a $V_{TFL}$ of 0.67 V, showing the total trap density decreased from $1.05 \times 10^{16}$ cm$^{-3}$ in the control film to a lower value of $6.13 \times 10^{15}$ cm$^{-3}$ in the target film. The TRPL results (Supplementary Fig. 15 and Supplementary Table 3) show that the target film exhibits a longer average PL lifetime of 693.34 ns while that of the pristine film is only 292.44 ns. Trap density of states (tDOS) in Fig. 4d shows that the target device possesses a lower tDOS in both the shallow-trap region (0.36–0.40 eV) and the deep-trap region (0.40–0.46 eV)[39,40]. These results confirm that moisture treatment is advantageous for obtaining a high-quality perovskite film with a much-reduced defect state.

We also studied the origin of the lower defect density of perovskite films after moisture treatment. As shown in Supplementary Fig. 16, we prepared the free-stand films by lifting off the wet intermediate perovskite films from the substrate[41]. The resultant sample was used to perform thermogravimetric analysis (TGA) at a constant temperature of 150 °C to simulate the annealing process of wet film. Compared with a mass loss of 0.79% of the control film, the target film exhibits a lower mass loss ratio of 0.45% (Supplementary Fig. 17). This lower mass loss can be attributed to the existence of hydrogen bonds between $H_2O$ and MA in humidified intermediate film, which can suppress the escape of MA cation during the annealing process. Therefore, high-quality perovskite films with suppressed cation vacancies can be achieved.

The improved film quality and reduced defect state are beneficial for improving the carrier transfer and mitigating the non-radiative recombination in the PSCs. We evaluated the built-in potential ($V_{bi}$) of the devices via the Mott-Schottky analysis (Fig. 4e). Compared with a built-in voltage ($V_{bi}$) of 0.81 V for the control device, the target device exhibits a higher $V_{bi}$ of 0.89 V, indicating a stronger internal driving force for separating the photo-generated carriers[42]. As shown in Fig. 4f, transient photovoltage measurements (TPV) of the full solar cell devices reveal that the photovoltage decay lifetime of the target device (1.42 ms) is much longer than that of the control device (0.39 ms), suggesting much-suppressed recombination in the target device[43].

### Enhancing the device performances by moisture treatment

To investigate the effect of moisture treatment on the photovoltaic performance of PSCs, we evaluated their J-V curves (Fig. 5a, b). The control device exhibits a reverse PCE (scanned from the open circuit to the short circuit) of 22.97% (with $V_{OC}$ of 1.13 V, $J_{SC}$ of 25.19 mA cm$^{-2}$, and FF of 80.40%) and a lower forward PCE of 22.16% (scanned from the short circuit to the open circuit), indicating a distinct hysteresis effect. The target device displays a higher impressive reverse PCE of 23.93% (with $V_{OC}$ of 1.15 V, $J_{SC}$ of 25.67 mA cm$^{-2}$, and FF of 80.86%) with the negligible hysteresis effect (the forward PCE is 23.45%). Figure 5c

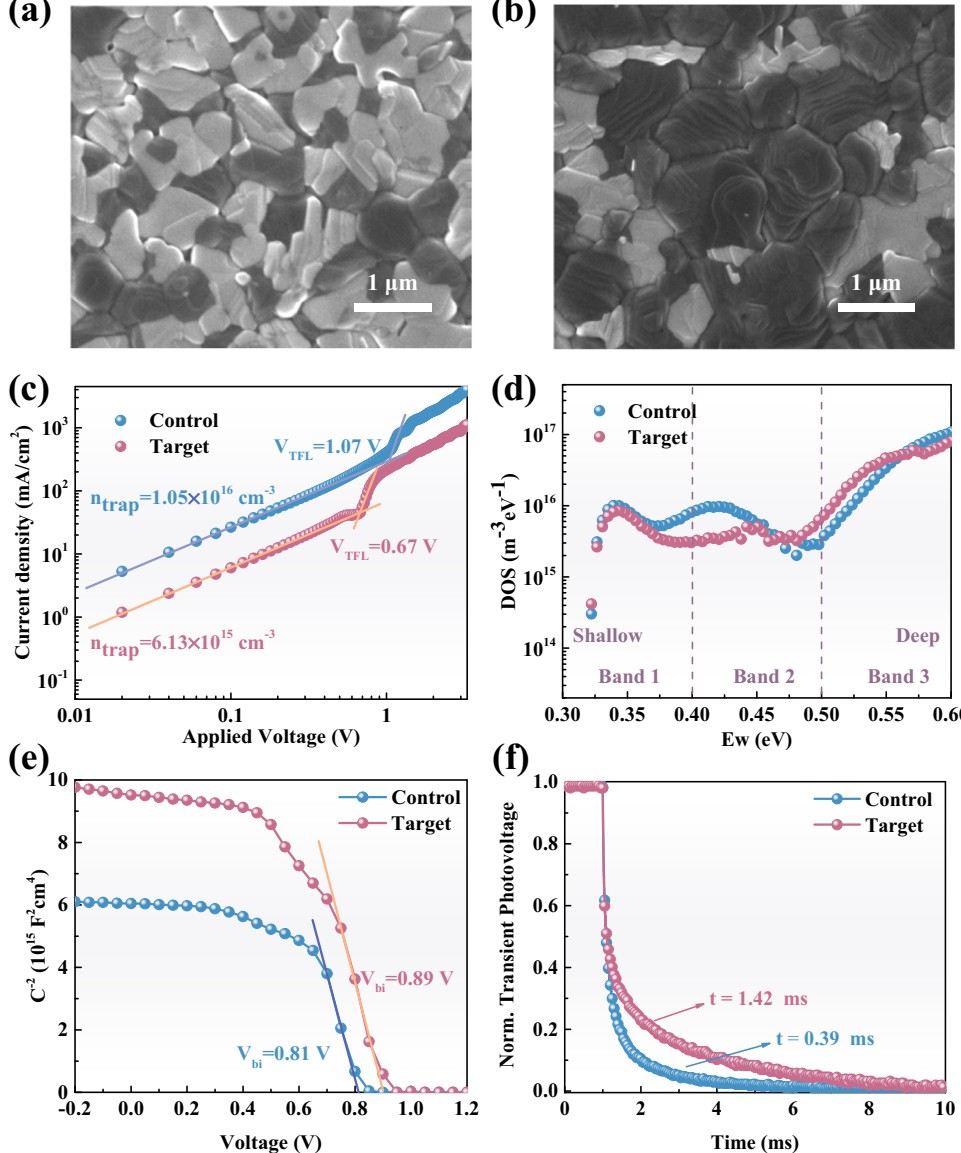

**Fig. 4 | The effect of moisture treatment on the properties of the final perovskite film and the carrier transport of the complete solar cells. a, b** Top-view SEM images of the annealed control and target perovskite films. **c** SCLC measured from the electron-only device with the structure of Glass/ITO/SnO$_2$/perovskite/ PC$_{60}$BM/Au. **d**. Trap density of states (tDOS) for the control and target devices with the structure of Glass/ITO/SnO$_2$/perovskite/Spiro-OMeTAD/Au. **e** Mott−Schottky plots and **f** TPV measurements of the control device and the target device. 'Normalized' is denoted as 'Norm'.

reveals that the integrated $J_{SC}$ values (24.31 mA cm$^{-2}$ for control and 24.92 mA cm$^{-2}$ for target) from the IPCE spectra are in good consistent with the $J_{SC}$ extracted from $J$-$V$ curves, indicating the reliability of the $J$-$V$ tests. Statistical distribution of the PCEs (Supplementary Fig. 18) shows that the overall efficiency of the target devices is much higher than that of the control.

To investigate the origin of the performance enhancement, we performed a series of measurements on the device level. Transient photocurrent (TPC) of the control and the target devices (Supplementary Fig. 19) show that their carrier lifetime is 2.05 and 2.08 μs, respectively, indicating similar charge separation and extraction of the photo-generated carriers in both devices at the short circuit[44]. We performed light intensity-dependent $J_{SC}$ and $V_{OC}$ measurements to study the charge separation and recombination within the PSCs. The relation between $J_{SC}$ and illumination intensities obeys the expression of $J_{SC} \propto I^\alpha$, where $I$ is the light intensity and α is the pre-exponential factor related to the bimolecular recombination[45]. The control device shows an α value of 0.956, whereas the target device displays a higher α

of 0.963 (Supplementary Fig. 20), suggesting the bimolecular recombination in the target device is suppressed. Moreover, we obtain the ideality factor $n$ of 1.53 and 1.21 for the control and the target devices, respectively (Fig. 5d). The lower ideal factor value of the target device confirms the reduction of the trap-associated Shockley−Read−Hall (SRH) recombination[46]. $J$-$V$ curves of PSCs under the dark condition show that the target device has a lower leakage current (Supplementary Fig. 21), indicating a better connection between functional layers and a higher rectification ratio, promoting the transport of charge carriers in the complete device[47].

We also evaluated the effect of moisture treatment on the stability of PSCs. As shown in Supplementary Fig. 22, the control device displays an evident decline in PCE at 0.97 V (the initial efficiency is 22.25%) in the first a few seconds of the measurement, while the target device displayed a stabilized PCE of 23.30% at 0.98 V even measuring for 600 s. For the shelf stability, statistical data (Supplementary Fig. 23) show that the PCE of target devices under all of the three storage conditions (unencapsulated and stored in the air; encapsulated and

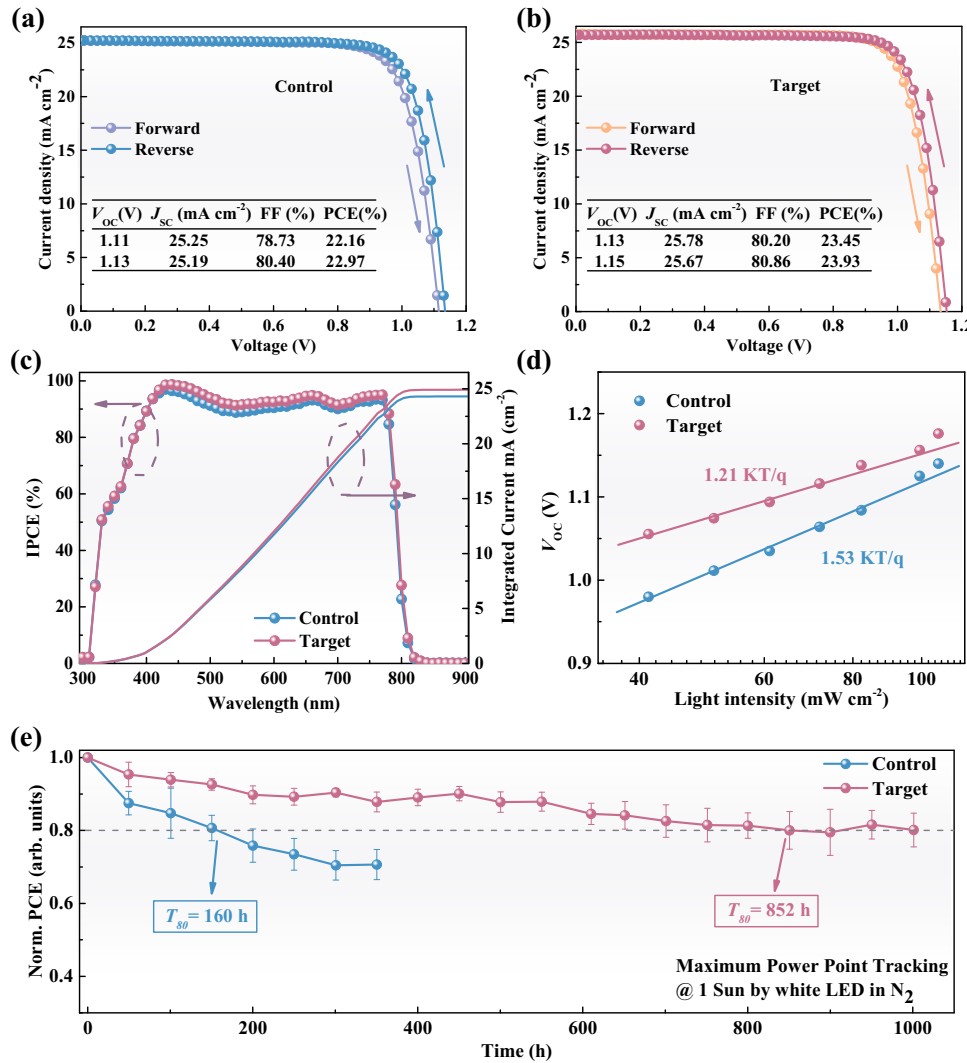

**Fig. 5 | Photovoltaic characterization of the devices based on the control and target perovskite films. a**, **b** $J$-$V$ curves and photovoltaic metrics under reverse and forward scans. **c** IPCE spectra and the corresponding integrated $J_{SC}$ of the control and the target PSCs. **d** Dependency of $V_{OC}$ as a function of the incident light intensity. **e** Statistic operational stability of the unencapsulated control and target devices which are tested under the maximum power point at 1-sun illumination in an $N_2$-filled glove box. All the error bars represent the standard deviation for six devices. 'Normalized' is denoted as 'Norm'.

stored in the air; unencapsulated and stored in $N_2$) show slightly better stability than the control devices. Additionally, the target devices under all three storage conditions show similar stability, implying that the 10%-humidity (during device storage) has little effect on the shelf stability. Figure 5e demonstrates the statistical operational stability which is tracked at the maximum power point under continuous 1-sun illumination in the $N_2$ atmosphere. We utilize $T_{80}$ (the time when the PCE declined to 80% of its initial efficiency) to evaluate the operational lifetime[48]. It can be observed that the average $T_{80}$ of the control device is only 160 h, while the target device shows a longer average $T_{80}$ of 852 h. Moreover, the champion stability of the target device exhibits a $T_{80}$ lifetime of about 1210 h while the $T_{80}$ of the champion stability of the control device is 275 h (Supplementary Fig. 24). The decline in PCE for both the control and the target devices should be attributed to the ion migration (caused by the influence of voltage bias and light)[49–51] and the loss of volatile components in the doped Spiro-OMeTAD layer[52]. The mobile charged ions will accumulate at the interfaces, resulting in worse interfacial band alignment and interfacial degradation. Therefore, defective perovskite solar cells exhibit a faster performance drop[53]. In addition, the loss of mobile components in the doped Spiro-OMeTAD will result in inferior hole extraction and

transport. Optimizing the HTL is expected to further improve the operational stability of PSCs.

## Discussion

In summary, we have developed a series of $N_2$-protected characterization techniques to study the actual effect of moisture on the perovskite films and PSCs. We find that water molecules can facilitate the reaction between organic cations and inorganic $PbI_2$ frame, trigger the fast-crystallization of the wet intermediate perovskite film before annealing, result in high-quality perovskite film with larger grains, higher crystallinity, and fewer defects. Finally, we obtained a PCE of approaching 24% (23.93%) with improved $V_{OC}$ and FF. Moreover, the corresponding device can remain over 80% of initial PCE after continuous 1-sun illumination of over 1200 h, showing robust operational stability. This work gives a feasible method for investigating metastable substances under protective conditions.

## Methods
### Materials
Unless otherwise stated, all chemicals were purchased from Sigma–Aldrich and used as received.

## Film deposition and device fabrication

The pre-patterned ITO glasses (15 Ω sq$^{-1}$) were sequentially washed by ultrasonic treatment in distilled water, acetone, isopropyl alcohol, and ethanol for 20 min, respectively. The surface wettability of the substrates was improved by a plasma treatment process (Harrick, PDC-002-HP) for 5 min. $SnO_2$ film was obtained by spin-coating the $SnO_2$ nanoparticle precursor (Alfa Aesar, 15% in $H_2O$ colloidal dispersion, diluted to 5%) onto ITO substrates (4000 rpm for 20 s) and then annealing in the ambient air (150 °C for 30 min). Another plasma treatment of 5 min was operated to clean the $SnO_2$ film surface after the ITO/$SnO_2$ substrates cooled down to room temperature. Thereafter, the ITO/$SnO_2$ substrates were transferred into an $N_2$-filled glove box for depositing the perovskite film. Here, perovskite films were fabricated via the sequential two-step deposition process. Firstly, $PbI_2$ films were obtained by spin-coating the $PbI_2$ solution onto ITO/$SnO_2$ substrates (2000 rpm for 30 s) and annealing at 70 °C for 1 min. The $PbI_2$ solution was prepared by dissolving $PbI_2$ (691.5 mg, TCI) and a 5% mole ratio of CsI (relative to $PbI_2$, 19.5 mg) in 1 mL mixed solvent of DMF (N, N-dimethylformamide) and DMSO (dimethylsulfoxide) with a v/v of 9:1. Subsequently, the organic salts solution was spin-coated on the obtained $PbI_2$ film at 1700 rpm for 30 s. The composition of organic salt solution was optimized[43,54–56]. And it was prepared by dissolving FAI (118.6 mg, Dyesol), MACl (18 mg, Dyesol), MABr (5.6 mg, Dyesol), and MAI (10 mg, Dyesol) in 2 mL of IPA.

For the moisture treatment, the samples were exposed to the air with a relative humidity of about 35 ± 5% for a controllable duration. Then the humidified samples were annealed on a hot plate at 150 °C for 15 min in the ambient air to obtain the target perovskite film. In contrast, the control samples were directly annealed at 150 °C for 15 min in $N_2$ ($H_2O$-0.1 ppm) to obtain the control perovskite film.

Before depositing the hole transport layers, the hexylamine hydrobromide (1.9 mg mL$^{-1}$, dissolved in chloroform) was dynamically spin-coated at 6000 rpm for 30 s to passivate the annealed perovskite films for both the control and the target samples. Hole transport layers were obtained by spin-coating (4000 rpm for 30 s) the 2,2′, 7,7′-Tetrakis [N,N-di(4-methoxyphenyl)amino]−9, 9′-spirobifluorene (Spiro-OMeTAD) onto the perovskite film without further annealing. The Spiro-OMeTAD precursor was prepared by dissolving 90 mg of Spiro-OMeTAD into 1 mL of chlorobenzene. In addition, 4-tert-butylpyridine (28.8 μL), and bis(trifluoromethane)sulfonimide lithium salt (17.5 μL, 520 mg mL-1 in acetonitrile) were incorporated to improve its conductivity. Finally, a layer of Au electrode (70 nm) was thermally evaporated on the Spiro-OMeTAD layer to finish the device fabrication.

## Characterization of perovskite films

To measure the actual information of as-prepared intermediate perovskite film, three basic steps were performed, including (1) fabricating the samples; (2) sealing the film with the proper cabins in $N_2$; (3) transferring the sample to the characterization equipment and recording the data. These processes guarantee that the wet intermediate perovskite film remains in its real original state without being influenced by the surroundings. The sample treatment in this work contains four stages, including (a) pristine (just completed the spin-coating process in the $N_2$-filled glove-box); (b) humidified (exposing the wet film in the ambient air with a ~35% relative humidity for about 60 s; (c) over humidified (prolonging the exposure time to more than 3 mins), and (d) annealed (putting the intermediate perovskite film on a hot plate at 150 °C for 15 min).

## UV-vis and PL characterization

Ultraviolet-visible absorption spectra (UV-vis) and steady-state photoluminescence (PL) spectra of different films were recorded with the home-assembled setup (Ocean Optics), which was installed in an $N_2$-filled glove box. For the characterization of the humidified samples, the films were held in the air with 35% R.H. for different times and then

taken into the $N_2$-filled glove box for UV-vis tests. The absorption spectra of the annealed perovskite film were also recorded in $N_2$.

## XPS characterization

XPS was recorded on the Thermo Fisher Scientific K-alpha+ system, using the monochrome Al Kα source with an energy of 1486.68 eV. The intermediate perovskite films were sputtered step by step by an Ar$^+$ ion beam with sputtering energy of 1000 eV. The sputtering rate was determined to be 0.2 nm s$^{-1}$ for tantalum pentoxide. This means that the sputter thickness is 0, 5, 10, 30, 70, and 110 nm for the sputtering duration of 0, 25, 50, 150, 350, and 550 s, respectively. However, the thickness doesn't represent the accurate depth of intermediate perovskite films due to different samples. To obtain the actual information of the samples before and after moisture treatment, a sealable cabin was used to transfer the samples from the $N_2$-filled glove box to the XPS equipment. The samples of FAI and MAI films were used to confirm the origin of the C = N and C-N peaks in XPS data.

## SEM characterization

Surface morphology and cross-sectional morphology of all samples were observed with a field-emission SEM (JEOL, JSM-7610F). Morphology changes of the same sample in different stages were measured via the above-mentioned methods of transferring samples. The as-prepared wet intermediate perovskite film was protected with a sealable shuttle cabin from the glove box to the SEM equipment. The annealed perovskite films were also transferred by the sealable shuttle cabin.

## XRD characterization

XRD was recorded with a SmartLab X-ray diffractometer (Rigaku Corporation) using a Cu Kα radiation source. XRD patterns for both the intermediate perovskite films and the annealed perovskite films were recorded with a sealable protection cabin. As the used protective tape in the specimen holder can generate an XRD peak at about 25°, Glass/ITO/$SnO_2$ substrate was used as a reference to remove the background.

## NMR characterization

$^1$H-NMR (500 MHz) were tested on Bruker Avance III 500 MHz NMR. The pristine and humidified wet intermediate perovskite films were re-dissolved from glass substrates by DMSO-d6. And then the solution was transferred into the nuclear magnetic tube for the semi-quantitative analysis of the relative amount of $H_2O$. Here, except for exposing the intermediate perovskite film in the air for moisture treatment, all processes were performed in an $N_2$-filled glove box ($H_2O$ < 0.1 ppm). Meanwhile, most of the $H_2O$ in the original DMSO-d6 were removed by molecular sieve before use. The ratio between the integrated area of the characteristic peaks between $H_2O$ and DMSO-d6 was used to indicate the relative water molecules amount. In addition, $H_2O$ in DMSO-d6 was difficult to be removed completely, and the increased $H_2O$ amount in the pristine sample can be ascribed to that introduced by the IPA solvent.

## TGA characterization

TGA system (DTG-60H) was used to analyze the mass loss of the intermediate perovskite film during the annealing process of 150 °C. To simulate the annealing process of the intermediate perovskite film with the TGA system, the wet intermediate perovskite film was lifted off from the substrate. Firstly, intermediate perovskite films were deposited on the poly[bis(4-phenyl) (2,5,6-trimethylphenyl) amine (PTAA) substrate. Then the pristine film or humidified films were immersed into chlorobenzene (CB) for about 10 min. PTAA, acting as a sacrificial layer, was dissolved by CB resulting in a free-stand upper intermediate perovskite layer. These samples were lifted off and floated on the solvent surface. Finally, the free-standing intermediate

perovskite film was loaded on the TGA balance to evaluate the mass loss during the annealing process. The sample was heated at a ramp rate of 5 °C min⁻¹ and then kept at 150 °C for 30 min. Note that the samples were transferred with a sealable cabin and the whole measurement process was performed in $N_2$ atmosphere. Meanwhile, the residual CB in the samples can be removed in the heating process from 70 °C to 150 °C. Finally, AFM images of the annealed perovskite films for the control and the target samples were recorded by using a Multimode 8 SPM system (Bruker).

## Device characterizations

Photovoltaic performances of the PSCs, including *J-V* curves and dark *J-V* curves (scanning rate of 500 mV s⁻¹, delay time of 40 ms, and voltage step of 20 mV.), were measured with a digital source meter (Keithley 2400) in a glove box filled with $N_2$. The light source of the *J-V* measurements was provided by the AAA solar simulator (Enli tech). The light intensity was calibrated with an NREL-calibrated Si solar cell with a Schott KG-5 filter. The mismatch factor between the reference cell and the device under test was calculated to be 1.002. Before the *J-V* test, no preconditioning such as light illumination was applied to the devices. The area of the PSCs was 0.2 cm², and the effective area of the devices was determined with a shadow mask (0.12 cm²). IPCE data were measured using a QE-R666 system (Enli tech) in the DC mode. The shelf stability of the unencapsulated device, stored in a drying box with a constant temperature of 25 °C and relative humidity of ~10%, was obtained by periodically testing the *J-V* measurements of the devices. The operational stability of the best-performed PSCs at the maximum power point (MPP) conditions was studied on a solar cell stability test system (Suzhou D&R Instruments Co., Ltd.) under 100 mW cm⁻² illumination. The light source was supplied with a white LED lamp. The temperature of the sample was about 55 °C.

## Reporting summary

Further information on research design is available in the Nature Research Reporting Summary linked to this article.

# Data availability

The data that support the findings of this study are available in the following repository: https://doi.org/10.6084/m9.figshare.20335185. The source data is provided with this work.

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

## Acknowledgements

This work was financially supported by the National Natural Science Foundation of China (22179042, U21A2078, and 51902110), the Natural Science Foundation of Fujian Province (2020J06021, and 2020J01064), and the Promotion Program for Young and Middle-aged Teacher in Science and Technology Research of Huaqiao University (ZQN-PY607, ZQN-806). We thank the comprehensive experiment center at Huaqiao University for providing the various test.

## Author contributions

Z.W. and L.X. supervised the work. Z.W. conceived the idea. Z.W., L.X., K.L., Y.Lu., and Y.J. fabricated devices and analyzed the data. T.L. and Z.L. performed tDOS measurements and analyzed the data. K.L., Y.Li., and C.T. performed NMR measurements and analyzed the data. K.L., L.Y., and P.S. performed TPV and TPC measurements. K.L. and L.X. co-wrote the paper. Z.W., L.X., K.L., and Y.Lu. revised the paper. All authors read and commented on the paper.

## Competing interests

The authors declare no competing interests.
