## [Peer Review File · Nature Communications]

Title: Moisture-triggered Fast Crystallization Enables Efficient and Stable Perovskite Solar CellsREVIEWER COMMENTS

Reviewer #1 (Remarks to the Author):

This work reveals the effect of moisture on perovskite and PSC device by using various N₂-protected characterization techniques and find a controllable moisture treatment for the intermediate perovskite can prepare high-quality perovskite films resulting in high performance PSC devices. It will greatly contribute to the development of high performance PSCs. Both the main body and supporting materials are well organized and properly written. It can be accepted by NC after minor revision.

1. Why the organic salt composition was set as 118.6 mg FAI, 18 mg MAI, 5.6 mg MABr, and 10 mg MAI in this study?

2. To evaluate the effect of moisture treatment on the PV performance (J-V curve) and the stability of PSCs, the unencapsulated device was stored in a drying box (the relative humidity is ~10%) in the air for 60 days. In fact, Two kinds of moisture treatments during perovskite film deposition (35% humidity) and during device characterization (10% humidity) were conducted, which will affect the performance and stability of devices. To distinguish these two effects of moisture, it is suggested to further investigate the stability of PSCs when the devices based on the perovskite with moisture treatment are encapsulated w/o moisture treatment.

3. The stability evaluation has revealed that both the control device and target device display a decline in PCE, and it need explain the reason of performance degradation.

Reviewer #2 (Remarks to the Author):

This paper by Liu et al. demonstrates improved efficiency and stability of perovskite solar cells and deciphers the role of moisture treatment in accelerating the formation of perovskite crystallization sites. However, processing under elevated relative humidity has both been reported and discussed extensively in the literature (not cited by the authors) over more than 5 years. For instance,

1. Pathak, S., Sepe, A., Sadhanala, A. et al. Atmospheric Influence upon Crystallization and Electronic Disorder and Its Impact on the Photophysical Properties of Organic-Inorganic Perovskite Solar Cells. *ACS Nano* 2015 9 (3), 2311-2320. DOI: 10.1021/nn506465n

2. Hu, Q., Zhao, L., Wu, J. et al. In situ dynamic observations of perovskite crystallization and microstructure evolution intermediated from [PbI₆]⁴⁻ cage nanoparticles. *Nat Commun* 8, 15688 (2017). <https://doi.org/10.1038/ncomms15688>

3. Yin, X. T., Guo, Y. X., Liu, J., Chen, P., Chen, W., Que, M. D., Que, W. X., Niu, C. M., Bian, J. H. and Yang, Y. D. (2017). Moisture annealing effect on CH₃NH₃PbI₃ films deposited by solvent engineering method. *Thin Solid Films* 636, 664-670.

4. Zhang, B., Zhang, M. J., Pang, S. P., Huang, C. S., Zhou, Z. M., Wang, D., Wang, N., and Cui, G. L. (2016). Carrier Transport in CH₃NH₃PbI₃ Films with Different Thickness for Perovskite Solar Cells. *Adv. Mater. Interfaces* 3, 1600327.

5. Chiang, C. H., and Wu, C. G. (2016). Film Grain-Size Related Long-Term Stability of Inverted Perovskite Solar Cells. *ChemSusChem* 9, 2666-2672.

6. Huang, H.-H., Ma, Z., Strzalka, J. et al. Mild water intake orients crystal formation imparting high tolerance on unencapsulated halide perovskite solar cells. *Cell Reports Physical Science*, 2021, 2, 100395. <https://doi.org/10.1016/j.xcrp.2021.100395>.

7. Lin, J., Lai, M., Dou, L. et al. Thermochromic halide perovskite solar cells. *Nature Mater* 17, 261–267 (2018). <https://doi.org/10.1038/s41563-017-0006-0>

As a result, I don't think the results presented here are new conceptual advances but a refinement of previous observations and protocols. I feel sure that the in-situ spectroscopy of the crystallization of perovskite films exposed to moisture will be useful to further clarify the impact of environmental conditions on the synthesis and stability of these films, and as such it will be of interest to other specialists working on these materials. Therefore, I would instead consider the manuscript more suited for a specialized journal.

Reviewer #3 (Remarks to the Author):

The authors focus on investigating the behavior of mixed perovskite (contains: Pb, I, Cs, FA, MA, Cl, Br) films without and with humidity exposure during their fabrication stage. The chosen measurement techniques (including UV-vis, PL, XPS, XRD, SEM, NMR, TGA) and methods aim for investigating the differences between the samples at different stages of preparation as directly as possible. The humidity treatment is detected to lead to higher formation of perovskite crystals, larger crystals with more vertically organized crystal boundaries, and in general more homogeneous films at all stages of film preparation. Full device analysis is under less focus in this work, but finally, full perovskite solar cells are also prepared from the films. The champion fresh device efficiency is 23.69% with humidity treatment vs. 22.06% without humidity treatment (confirmed also statistically in the analysis of the whole sample batch), and T80 stability is 803h under visible light only maximum power point tracking for the solar cell with humidity treatment vs. 182h without the treatment (shown only to one sample so this is not a statistically confirmed result).

The topic area of the work is timely and interesting, since there are contradictory results in literature related to the effects of humidity on perovskite performance (especially in terms of stability performance, and for mixed perovskites in contrast to e.g. simple MAPI perovskite) and better understanding the effects would allow a more purposeful design of more usable materials for various applications such as perovskite solar cells (handled here) and LEDs. Due to the wide application area of the results of this work, it should be of interest to the broad readership of the *Nature Communications*. Furthermore, I find the approach of targeting direct and almost in-situ evidence instead of secondary measurement methods (e.g. device performance or properties of the films significantly after preparation time) an inspiring example that should also motivate similar work in other tricky areas of materials research. The manuscript is clear and concise, the conclusions supported by the results, and the experimental work seems to have been implemented in an exceptionally meticulous way.

My major concerns are listed as follows:

- Perovskite composition. The investigated perovskite composition is not directly stated anywhere in the manuscript even though it can be deduced from the sample preparation part of Methods section. This is essential information that should be stated clearly, possibly already in the abstract.

- Stability analysis. The stability results of full perovskite solar cells shown here (shelf life and maximum power point tracking) are not statistically significant since only one champion device is shown from each group. The variability in the efficiency performance of perovskite solar cells is not negligible (as shown in the supplementary figure 17 showing fresh device efficiencies; the least efficient cell is ~19.5% and the most efficient is 23.69%), which suggests the variability in the stability performance is not negligible, either. Showing just the champion devices does not allow for quantitative estimation of the difference in stability (although I admit that the difference between 803h and 182h of T80 seems large enough that one can regard the stability improvement as having been qualitatively proved). The authors should preferably provide statistically relevant amount of data on device stability, or at least clearly state that the full device stability results are single-cell data and thus cannot quantitatively represent the difference between the two groups of devices.

- On a similar note, the number of perovskite film samples prepared and measured in this work is not stated clearly in the manuscript. Was it one film that was exposed to all the measurements? Or one film/multiple films for each measurement technique? What is the general repeatability/variability of the film properties using the described sample preparation process in your laboratory? Please clarify these questions.

I cannot recommend the publication of this manuscript unless the major concerns stated above are adequately considered.

Additionally, I have the following minor comments on the work:

- I did not realize based on the title + abstract that the authors aim for direct measurements vs. secondary measurements of the differences in the films. Please consider stating this more clearly already early in the paper, since to me it is the most interesting aspect of the work.

- Some of the measurements seem to have been performed practically in-situ during the film preparation and solidification, some have required transporting the samples in the specifically prepared protective containers. Also solar cell preparation requires additional stages of work which takes time. Currently it is not clear how similar/different are the storage times under N₂ between the different measurements performed to films at the same preparation stage. Please comment on this.

- Supplementary Figure 9: Is the sample currently labeled as "Annealed" actually "Over humidified and annealed"?

Reviewer #1

This work reveals the effect of moisture on perovskite and PSC device by using various N₂-protected characterization techniques and find a controllable moisture treatment for the intermediate perovskite can prepare high-quality perovskite films resulting in high performance PSC devices. It will greatly contribute to the development of high performance PSCs. Both the main body and supporting materials are well organized and properly written. It can be accepted by NC after minor revision.

General response: We appreciate reviewer #1 for reviewing our manuscript and providing constructive comments to improve the work. We carefully considered these comments, and the detailed response can be found in the point-to-point response below.

Comment 1. Why the organic salt composition was set as 118.6 mg FAI, 18 mg MACl, 5.6 mg MABr, and 10 mg MAI in this study?

Response: Thanks for the reviewer's comment. The design principles of the organic salt composition are as follows: 1) The concentration of FA is determined by the lowest concentration that ensures the complete reaction between organic salt and the PbI₂ film in the sequential deposition (Shen, L. *et al.*, *J Mater. Chem. A*, **2021**, 9, 20807). 2) MABr and MAI are used for the doping of MA cation and Br⁻ anion to suppress the spontaneous α -to- δ phase transition of the FAPbI₃-based perovskite (Li N., *et al.*, *Nat. Energy*, **2019**, 4, 408). The doping amount of MA and Br⁻ is expected to be as low as possible to avoid doping-induced bandgap enlargement. Because a larger bandgap delivers lower theoretical device efficiency. 3) The content of MACl was adjusted to obtain perovskite film with large crystal grains (M. Kim, *et al.*, *Joule*, **2019**, 3, 2179; Ye, F. *et al. Adv. Mater.*, **2021**, 33, 2007126). After optimization based on these concerns in our lab, the organic salt solution was set as dissolving 118.6 mg FAI, 18 mg MACl, 5.6 mg MABr, and 10 mg MAI in 2 mL of IPA. The corresponding references (Ref. 43, 54-56 in the revised manuscript) are cited in the revised manuscript.

Comment 2. To evaluate the effect of moisture treatment on the PV performance (J-V curve) and the stability of PSCs, the unencapsulated device was stored in a drying box (the relative humidity is ~10%) in the air for 60 days. In fact, two kinds of moisture treatments during perovskite film deposition (35% humidity) and during device characterization (10% humidity) were conducted, which will affect the performance and stability of devices. To distinguish these two effects of moisture, it is suggested to further investigate the stability of PSCs when the devices based on the perovskite with moisture treatment are encapsulated w/o moisture treatment.

Response: Thanks for the reviewer's important suggestion. We agree with the reviewer that the effect of the two kinds of moisture treatments, during perovskite film deposition (35% humidity) and device characterization (10% humidity), should be distinguished. As the long-term operational stability test was done in the N₂-filled glovebox (without the interference of the 10%-humidity treatment), we focused solely on the evaluation of shelf stability under different conditions to understand the effect of the 10% humidity during device storage. We measured the *J-V* curves of the control and target devices under three conditions: 1) unencapsulated devices stored in the air with 10% humidity; 2) encapsulated devices stored in the air with 10% humidity; 3) unencapsulated devices stored in the N₂-filled glovebox. Condition 3 is used to crosscheck the results of condition 2 for devices prepared in 35% humidity and stored without moisture. The statistical results are presented in Supplementary Fig. 23. It can be seen that the PCE of the target devices under all of the three storage conditions show similar stability, which can keep about 90% of their initial PCE after over 30 days. In addition, the shelf stability of the control devices is slightly inferior to the target. These results suggest that the effect of the 10%-humidity (during device storage) on the shelf stability is negligible. And the main effect originates from the 35% humidity treatment during film fabrication.

Supplementary Figure 23 | Shelf stability of the control and target devices under different storage conditions: 1) Unencapsulated and stored in the air; 2) Encapsulated and stored in the air; 3) Unencapsulated and stored in N₂. All the error bars represent the standard deviation for 10 devices, except for that of the control device with encapsulation representing 8 devices.

Changes in the manuscript: For the shelf stability, statistical data (Supplementary Figure 23) show that the PCE of target devices under all of the three storage conditions (unencapsulated and stored in the air; encapsulated and stored in the air; unencapsulated and stored in N₂) show slightly better stability than the control devices. Additionally, the target devices under all three storage conditions show similar stability, implying that the 10%-humidity (during device storage) has little effect on the shelf stability.

Comment 3. The stability evaluation has revealed that both the control device and target device display a decline in PCE, and it need explain the reason of performance degradation.

Response: Thanks for the reviewer's comment. The decline in PCE should be attributed to the ion migration caused by the influence of voltage bias and light (Li, N. et al. *Chem. Soc. Rev.*, **2020**, 49, 823; Azpiroz, J. M. et al. *Energy Environ. Sci.*, **2015**, 8, 2118; Yoon, S. J. et al. *ACS Energy Lett.*, **2016**, 1, 290), and the loss of volatile components in the doped Spiro-OMeTAD layer (Wang, L. et al. *J. Mater. Chem. A*, **2020**, 8, 14106). The mobile charged ions will accumulate at the interfaces, resulting in worse interfacial

band alignment and interfacial degradation. Therefore, defective perovskite solar cells exhibit a faster performance drop (Chen B., et al. *Chem. Soc. Rev.*, **2019**, 48, 3842). In addition, the loss of mobile components in the doped Spiro-OMeTAD will result in inferior hole extraction and transport. This also leads to the decline in PCE. The corresponding references (Ref. 49-53 in the revised manuscript) are cited in the revised manuscript.

Changes in the manuscript: The decline in PCE for both the control and the target devices should be attributed to the ion migration (caused by the influence of voltage bias and light) and the loss of volatile components in the doped Spiro-OMeTAD layer. The mobile charged ions will accumulate at the interfaces, resulting in worse interfacial band alignment and interfacial degradation. Therefore, defective perovskite solar cells exhibit a faster performance drop. In addition, the loss of mobile components in the doped Spiro-OMeTAD will result in inferior hole extraction and transport. Optimizing the HTL is expected to further improve the operational stability of PSCs.

Reviewer #2

This paper by Liu et al. demonstrates improved efficiency and stability of perovskite solar cells and deciphers the role of moisture treatment in accelerating the formation of perovskite crystallization sites. However, processing under elevated relative humidity has both been reported and discussed extensively in the literature (not cited by the authors) over more than 5 years. For instance,

1. Pathak, S., Sepe, A., Sadhanala, A. et al. Atmospheric Influence upon Crystallization and Electronic Disorder and Its Impact on the Photophysical Properties of Organic-Inorganic Perovskite Solar Cells. *ACS Nano* 2015 9 (3), 2311-2320. DOI: 10.1021/nm506465n
2. Hu, Q., Zhao, L., Wu, J. et al. In situ dynamic observations of perovskite crystallization and microstructure evolution intermediated from [PbI₆]⁴⁻ cage nanoparticles. *Nat Commun* 8, 15688 (2017). <https://doi.org/10.1038/ncomms15688>
3. Yin, X. T., Guo, Y. X., Liu, J., Chen, P., Chen, W., Que, M. D., Que, W. X., Niu, C. M., Bian, J. H. and Yang, Y. D. (2017). Moisture annealing effect on CH₃NH₃PbI₃ films deposited by solvent engineering method. *Thin Solid Films* 636, 664-670.
4. Zhang, B., Zhang, M. J., Pang, S. P., Huang, C. S., Zhou, Z. M., Wang, D., Wang, N., and Cui, G. L. (2016). Carrier Transport in CH₃NH₃PbI₃ Films with Different Thickness for Perovskite Solar Cells. *Adv. Mater. Interfaces* 3, 1600327.
5. Chiang, C. H., and Wu, C. G. (2016). Film Grain-Size Related Long-Term Stability of Inverted Perovskite Solar Cells. *ChemSusChem* 9, 2666-2672.
6. Huang, H.-H., Ma, Z., Strzalka, J. et al. Mild water intake orients crystal formation imparting high tolerance on unencapsulated halide perovskite solar cells. *Cell Reports Physical Science*, 2021, 2, 100395. <https://doi.org/10.1016/j.xcrp.2021.100395>.
7. Lin, J., Lai, M., Dou, L. et al. Thermochromic halide perovskite solar cells. *Nature Mater* 17, 261–267 (2018). <https://doi.org/10.1038/s41563-017-0006-0>

As a result, I don't think the results presented here are new conceptual advances but a refinement of previous observations and protocols. I feel sure that the in-situ

spectroscopy of the crystallization of perovskite films exposed to moisture will be useful to further clarify the impact of environmental conditions on the synthesis and stability of these films, and as such it will be of interest to other specialists working on these materials. Therefore, I would instead consider the manuscript more suited for a specialized journal.

Response: We thank the reviewer for the valuable comments to help us improve the quality of the manuscript. We regret that the original version of the manuscript does not present explicit information about the work's importance and novelty.

We agree that processing perovskite under elevated relative humidity is not a novel method. We further conducted more comprehensive literature research. After analyzing the previously reported works, we concluded that the current work presents important advances in the effect of moisture on the state-of-the-art formamidinium-based mixed-cation mixed-halide perovskite, which has received tremendous attention as an absorber in record-efficiency perovskite solar cells.

The reports listed by the reviewer are important for understanding the effect of moisture on the formation (#6), phase transition (#7), and photoelectric properties (#1 and #3) of perovskite. The other references (#2, #4, and #5) didn't directly study the effect of moisture on perovskite, although the devices in reference #2 were prepared in the ambient air. Among these reports, #6 by Huang et al. is especially important and is cited as ref. 29 in the revised manuscript. They captured the crystal formation process from the precursor to the final perovskite ($\text{MAPbI}_{3-x}\text{Cl}_x$). *In-situ* GIWAXS experiments were carried out during annealing the film under different relative humidity. They concluded that a moderate water content accelerates the crystal formation and enhances the texture-orientation of the film.

The representative reports in the literature investigating the effect of water (or moisture) on the perovskite materials with different compositions are summarized in Supplementary Table 1 of the revised SI. The perovskite composition, processing methods with water, research methods, efficiency and stability of the reported devices can be found in this table. Water was introduced via different processing methods (in the precursors, during spin-coating, during annealing, or post-treatment of the annealed

perovskite film). The advantages of moisture on perovskite were claimed to be that a suitable dose of water can promote perovskite crystal formation, enlarge the crystal grains, enhance the crystal orientation, enhance the diffusion of the ions, and passivate the defects, *etc.* The drawback of water on perovskite is that it may destroy the morphology of the film and cause degradation.

We note that the investigated effect of H₂O is always coupled with other factors such as the surrounding atmosphere, the heating thermal energy, and the solvent, which depends on the processing strategies. Decoupling these factors and isolating the actual effect of the water from them is essential to obtain the so-called ‘direct evidence’ of the water-induced effects. Moreover, the effect of water on perovskite materials depends on the perovskite composition (reference #3 suggested by the reviewer). Our work focuses on the FA-based perovskite with the composition of (Cs_{0.05}FA_{0.75}MA_{0.20})Pb(I_{0.96}Br_{0.04})₃, which possesses a low bromide/iodide ratio thus lowered bandgap and higher theoretical efficiency. Insights into this system may contribute considerably to the development of high-performance perovskite solar cells that simultaneously possess high efficiency and robust stability. Furthermore, the research methods and in-situ technologies reported in this work may allow a more purposeful design of environment-sensitive materials for other application areas thus as Sn-based perovskite solar cells and perovskite LED.

Regarding the ‘new conceptual advances’ of this work, we summarize the contribution of this work as follows. Firstly, direct experimental evidence reveals that water alone can trigger the fast crystallization of perovskite. Secondly, we emphasize that the major effect of water on the intermediate perovskite occurs within the timescale of minutes. Thirdly, direct evidence reveals that water help uniformizes the distribution of chemical species in the intermediate perovskite and this effect transforms into high-performance perovskite solar cells with improved operational stability.

In the revised manuscript, we have cited the related references and revised the abstract and introduction part correspondingly. We sincerely invite reconsider our work for publishing in *Nature Communications*.

Changes in the manuscript:

Change 1: Here, we develop various N₂-protected characterization techniques to comprehensively study the effect of moisture on the efficient cesium, methylammonium (MA), and formamidinium (FA) triple-cation perovskite (Cs_{0.05}FA_{0.75}MA_{0.20})Pb(I_{0.96}Br_{0.04})₃. In contrast to the secondary measurements, the established air-exposure-free techniques allow us directly monitor the influence of moisture during perovskite crystallization.

Change 2: Huang et al. utilized in-situ grazing-incident wide-angle X-ray scattering (GIWAXS) to capture the phase transformation of the CH₃NH₃I_xCl_{3-x} perovskite films during annealing under different relative humidity. They concluded that a moderate water content accelerates the crystal formation and enhances the texture-orientation of the film.

Change 3: It is noted that many previous reports have investigated the role of H₂O in improving the quality of perovskite films (summarized in Supplementary Table 1). However, the investigated effect of H₂O is always coupled with other factors such as the surrounding atmosphere, the heating thermal energy, and the solvent, which depends on the processing strategies. Moreover, most of the conclusions were even drawn from studying the final device performance, which is indirect evidence influenced by many complicated factors. Decoupling the actual role of moisture from these factors is essential to obtain the so-called ‘direct evidence’ of the water-induced effects. Systematic studies monitoring the intermediate perovskite treated by moisture alone are more convincing.

Change 4: Moreover, the effect of water depends on the perovskite composition. As the cesium, MA, and formamidinium (FA) triple-cation perovskite possessing a low bromide/iodide ratio delivers higher theoretical efficiency, gaining insights into the effect of moisture on this system may contribute to a more purposeful design of high-performance PSCs that simultaneously possess high efficiency and robust stability.

Reviewer #3

The authors focus on investigating the behavior of mixed perovskite (contains: Pb, I, Cs, FA, MA, Cl, Br) films without and with humidity exposure during their fabrication stage. The chosen measurement techniques (including UV-vis, PL, XPS, XRD, SEM, NMR, TGA) and methods aim for investigating the differences between the samples at different stages of preparation as directly as possible. The humidity treatment is detected to lead to higher formation of perovskite crystals, larger crystals with more vertically organized crystal boundaries, and in general more homogeneous films at all stages of film preparation. Full device analysis is under less focus in this work, but finally, full perovskite solar cells are also prepared from the films. The champion fresh device efficiency is 23.69% with humidity treatment vs. 22.06% without humidity treatment (confirmed also statistically in the analysis of the whole sample batch), and T80 stability is 803h under visible light only maximum power point tracking for the solar cell with humidity treatment vs. 182h without the treatment (shown only to one sample so this is not a statistically confirmed result).

The topic area of the work is timely and interesting, since there are contradictory results in literature related to the effects of humidity on perovskite performance (especially in terms of stability performance, and for mixed perovskites in contrast to e.g. simple MAPI perovskite) and better understanding the effects would allow a more purposeful design of more usable materials for various applications such as perovskite solar cells (handled here) and LEDs. Due to the wide application area of the results of this work, it should be of interest to the broad readership of the Nature Communications. Furthermore, I find the approach of targeting direct and almost in-situ evidence instead of secondary measurement methods (e.g. device performance or properties of the films significantly after preparation time) an inspiring example that should also motivate similar work in other tricky areas of materials research. The manuscript is clear and concise, the conclusions supported by the results, and the experimental work seems to have been implemented in an exceptionally meticulous way.

General response: We greatly thank the reviewer for the constructive comments.

My major concerns are listed as follows:

Comment 1. Perovskite composition. The investigated perovskite composition is not directly stated anywhere in the manuscript even though it can be deduced from the sample preparation part of Methods section. This is essential information that should be stated clearly, possibly already in the abstract.

Response: Thanks for the reviewer's important comment. The perovskite composition is $(\text{Cs}_{0.05}\text{FA}_{0.75}\text{MA}_{0.20})\text{Pb}(\text{I}_{0.96}\text{Br}_{0.04})_3$, which is stated in the abstract of the revised manuscript.

Although the perovskite composition can be conventionally deduced from the precursor, we tried our best to determine it more accurately. The ratio of Cs^+ in the A site is determined to be 0.05 as the Cs/Pb ratio is fixed as 0.05 in the precursor solution (CsI and PbI_2 dissolved in DMF/DMSO). The FA and MA ratios can't directly be accurately determined by the precursor in the organic salts solution because they compete with each other to enter the PbI_2 film. Therefore, we performed chemical element analysis to obtain the C/N ratio (determined to be about 0.48), then, the ratio of FA and MA is determined to be 0.75 and 0.20, respectively. Finally, the I/Br ratio is determined by comparing the UV-vis absorption spectra of the target perovskite film with the known $\text{Cs}_{0.05}\text{FA}_{0.75}\text{MA}_{0.20}\text{Pb}(\text{I}_{1-x}\text{Br}_x)_3$ perovskite and x is determined to be 0.04. It is reported that Cl escapes after annealing and there is no residual Cl residual in the final perovskite film (M. Kim, et al., *Joule*, **2019**, 3, 2179; Ye, F. et al. *Adv. Mater.*, **2021**, 33, 2007126). Finally, the perovskite composition in this work can be determined to be $(\text{Cs}_{0.05}\text{FA}_{0.75}\text{MA}_{0.20})\text{Pb}(\text{I}_{0.96}\text{Br}_{0.04})_3$.

Supplementary Figure 25 | UV-Vis spectra of the perovskite films with the composition of $(\text{Cs}_{0.05}\text{FA}_{0.75}\text{MA}_{0.20})\text{Pb}(\text{I}_{1-x}\text{Br}_x)_3$. The bromide percentage in the legend represents different molar ratios between MABr and MAI in the precursor of the one-step method.

Changes in the manuscript: Here, we develop various N_2 -protected characterization techniques to comprehensively study the effect of moisture on the efficient cesium, methylammonium (MA), and formamidinium (FA) triple-cation perovskite $(\text{Cs}_{0.05}\text{FA}_{0.75}\text{MA}_{0.20})\text{Pb}(\text{I}_{0.96}\text{Br}_{0.04})_3$.

Changes in the revised SI:

Supplementary Note 1. Investigation of the composition of the perovskite film

We adopted chemical element analysis, and UV-Vis spectra to investigate the composition of perovskite. First, the ratio of Cs^+ in the A site is determined to be 0.05 as the Cs/Pb ratio is fixed as 0.05 in the precursor solution (CsI and PbI_2 dissolved in DMF/DMSO). The FA and MA ratios can't directly be determined by the precursor in the organic salts solution because they compete with each other to enter the PbI_2 film. Therefore, we collected perovskite powder that was peeled from the perovskite films to investigate the chemical element composition (Vario EL Cube). The obtained C/N ratio was determined to be about 0.48. So, the ratio of FA and MA was determined to be 0.75 and 0.20, respectively. Next, we fabricated perovskite films with varied molar ratios of MABr/MAI and fixed Cs, FA, and MA ratios in the one-step method. Note that the

composition of the samples fabricated by the one-step method can be directly determined by the precursor. Finally, the I/Br ratio is determined by comparing the UV-vis absorption spectra of the target perovskite film with the known $\text{Cs}_{0.05}\text{FA}_{0.75}\text{MA}_{0.20}\text{Pb}(\text{I}_{1-x}\text{Br}_x)_3$ perovskite and x is determined to be 0.04. It is reported that Cl escapes after annealing and there is no residual Cl in the final perovskite film. Finally, the perovskite composition in this work was determined to be $(\text{Cs}_{0.05}\text{FA}_{0.75}\text{MA}_{0.20})\text{Pb}(\text{I}_{0.96}\text{Br}_{0.04})_3$.

Comment 2. Stability analysis. The stability results of full perovskite solar cells shown here (shelf life and maximum power point tracking) are not statistically significant since only one champion device is shown from each group. The variability in the efficiency performance of perovskite solar cells is not negligible (as shown in the supplementary figure 17 showing fresh device efficiencies; the least efficient cell is ~19.5% and the most efficient is 23.69%), which suggests the variability in the stability performance is not negligible, either. Showing just the champion devices does not allow for quantitative estimation of the difference in stability (although I admit that the difference between 803h and 182h of T80 seems large enough that one can regard the stability improvement as having been qualitatively proved). The authors should preferably provide statistically relevant amount of data on device stability, or at least clearly state that the full device stability results are single-cell data and thus cannot quantitatively represent the difference between the two groups of devices.

Response: Thanks for this important comment. To confirm the effect of moisture treatment on device stability, we further conducted additional experiments and provided the statistical data.

1) Stability analysis

For the shelf stability, statistical data (Supplementary Figure 23) show that the PCE of the target devices under all of the three storage conditions (unencapsulated and stored in the air, encapsulated and stored in the air, and unencapsulated and stored in N₂) show slightly better stability than the control devices. Additionally, the target devices under all three storage conditions show similar stability, implying that the 10%-humidity (during device storage) has little effect on the shelf stability.

For the operational stability, Figure 5e shows that the average T_{80} lifetime of the control device is 160 h, while the target device shows a longer average value of 852 h. Additionally, the champion stability of the target device exhibits a T_{80} lifetime of about 1210 h while the T_{80} of the champion stability of the control device is 275 h (Supplementary Fig. 24). These results statistically confirm the effects of moisture treatment on improving the stability of PSCs.

Supplementary Figure 23 | Shelf stability of the control and target devices under different storage conditions: 1) Unencapsulated and stored in the air; 2) Encapsulated and stored in the air; 3) Unencapsulated and stored in N_2 . All the error bars represent the standard deviation for 10 devices, except for that of the control device with encapsulation representing 8 devices.

Fig. 5(e) Statistic operational stability of the unencapsulated control and target devices which are tested under the maximum power point at 1-sun illumination in an N_2 -filled glove box. All the error bars represent the standard deviation for six devices.

Supplementary Figure 24 | The operational stability of the unencapsulated champion device of the control and target samples which are tested under the maximum power point at 1-sun illumination in an N_2 -filled glove box.

Changes in the manuscript: For the shelf stability, statistical data (Supplementary Figure 23) show that the PCE of target devices under all of the three storage conditions (unencapsulated and stored in the air; encapsulated and stored in the air, and unencapsulated and stored in N₂) show slightly better stability than the control devices. Additionally, the target devices under all three storage conditions show similar stability, implying that the 10%-humidity (during device storage) has little effect on the shelf stability. Fig. 5e demonstrates the statistical operational stability tracked at the maximum power point under continuous 1-sun illumination in the N₂ atmosphere. We utilize T_{80} (the time when the PCE declined to 80% of its initial efficiency) to evaluate the operational lifetime. It can be observed that the average T_{80} of the control device is only 160 h, while the target device shows a longer average T_{80} of 852 h. Moreover, the champion stability of the target device exhibits a T_{80} lifetime of about 1210 h, while the T_{80} of the champion stability of the control device is 275 h (Supplementary Fig. 24).

2) Photovoltaic performances

During fabricating additional solar cells for investigation of stability, the champion PCE was further updated to 23.93%. In the revised manuscript, the hysteresis, IPCE spectra, stabilized power output, and statistical PCE of the devices were also updated accordingly.

Fig. 5(a, b) J - V curves and photovoltaic metrics under reverse and forward scans.

Fig. 5(c) IPCE spectra and the corresponding integrated JSC of the control and the target PSCs.

Supplementary Figure 18 | Statistical PCE of the control and target devices.

Changes in the manuscript: The control device exhibits a reverse PCE (scanned from the open circuit to the short circuit) of 22.97% (with V_{OC} of 1.13 V, J_{SC} of 25.19 mA cm⁻², and FF of 80.40%) and a lower forward PCE of 22.16% (scanned from the short circuit to the open circuit), indicating a distinct hysteresis effect. The target device displays a higher impressive reverse PCE of 23.93% (with V_{OC} of 1.15 V, J_{SC} of 25.67 mA cm⁻², and FF of 80.86%) with the negligible hysteresis effect (the forward PCE is 23.45%). Fig. 5c reveals that the integrated J_{SC} values (24.31 mA cm⁻² for control and 24.92 mA cm⁻² for target) from the IPCE spectra are in good consistent with the J_{SC} extracted from $J-V$ curves, indicating the reliability of the $J-V$ tests. Statistical distribution of the PCEs (Supplementary Fig. 18) shows that the overall efficiency of the target devices is much higher than that of the control.

Supplementary Figure 22 | Steady-state current density measured at the maximum power point for the control and target devices.

Changes in the manuscript: We also evaluated the effect of moisture treatment on the stability of PSCs. As shown in Supplementary Fig. 22, the control device displays an evident decline in PCE at 0.97 V (the initial efficiency is 22.25%) in the first a few seconds of the measurement, while the target device displayed a stabilized PCE of 23.30% at 0.98 V even measuring for 600 s.

Comment 3. On a similar note, the number of perovskite film samples prepared and measured in this work is not stated clearly in the manuscript. Was it one film that was exposed to all the measurements? Or one film/multiple films for each measurement technique? What is the general repeatability/variability of the film properties using the described sample preparation process in your laboratory? Please clarify these questions.

Response: Thanks for the reviewer's comments. It is essential to state clearly the number of the samples and the history of each sample for each characterization technique. For PL, XPS, NMR, and TGA measurements, individual fresh samples (pristine and humidified) were prepared for each experiment. For SEM and XRD experiments, the perovskite films before (Pristine) and after (Humidified) moisture treatment was based on the same sample. After measuring the pristine one, the film was then treated with moisture and the humidified sample was measured. Specifically, for the measurement of UV-vis absorption as a function of time, one pristine and one target film was measured at different aging times. The corresponding sentences describing the measured samples are revised accordingly in the revised manuscript.

To demonstrate the repeatability of the effect of moisture treatment on film properties, we provided some complementary results for the corresponding characterizations.

1) Repeatability of the optical appearance

For the same intermediate perovskite film, we can also observe similar color changes when this film is exposed to moisture at different times. This result is consistent with that based on different samples at each stage.

Supplementary Figure 4 | Photographs of the same sample in different stages of pristine, humidified, over humidified wet intermediate perovskite film, and annealed perovskite film.

Changes in the manuscript: Besides, when the same wet intermediate perovskite film was exposed to the air, similar color changes were observed (Supplementary Fig. 4).

2) Repeatability of PL and UV-vis spectroscopy

The PL spectra excited from the front side and backside (Figure R1) and the evolution of the UV-Vis spectra as a function of aging time (Figure R2) are confirmed by another batch of samples.

Figure R1 | Steady-state photoluminescence spectra of another batch of samples that contains intermediate perovskite film w/o (Pristine) and with (Humidified) moisture treatment.

Figure R2 | The evolution of UV-Vis spectra of another batch of samples that contain intermediate perovskite films w/o (Pristine) and with (Humidified) moisture treatment.

3) Repeatability of morphology evolution

We recorded the morphology evolution of the intermediate perovskite films before (Pristine) and after (Humidified) moisture treatment from another batch of samples (Fig.

R3). It shows that the morphology changes from the messy grain distribution to more homogenous large grains, which can confirm the repeatability in film morphology induced by moisture treatment.

Figure R3 | Top-view SEM images of the same intermediate perovskite films before (Pristine) and after (Humidified) humidified treatment (The insets were enlarged from the corresponding pictures).

4) Repeatability of the crystal structure of the wet intermediate perovskite film

As shown in Figure R4, XRD results of another batch of the intermediate perovskite film confirm that moisture treatment can induce the crystallization of the intermediate perovskite films.

Figure R4 | XRD patterns from another batch of intermediate perovskite film before (Pristine) and after (Humidified) moisture treatment.

In addition, for the NMR and TGA measurements, the tested samples were obtained from many films other than only one film. This makes the results more reliable. Specially, we used DMSO-d₆ as a solvent to dissolve many wet perovskite films and collected the solution into the nuclear magnetic tubes. For TGA experiments, we also

lifted off many wet perovskite films and collected the sheets into the TGA balance.

Therefore, the reproducibility of the properties of perovskite films is confirmed by double-checked experiments. Moreover, during the fabrication of perovskite solar cells, we can observe repeatable phenomena with naked eyes, implying good repeatability of the effect of moisture treatment by controlled humidity.

Changes in the manuscript: It is noted that the reproducibility of the optical properties, film morphology, and crystal structure of the films is confirmed by double-checked experiments, implying good repeatability of the effect of moisture treatment by the controlled humidity.

Comment 4. I cannot recommend the publication of this manuscript unless the major concerns stated above are adequately considered.

Response: Thanks so much for the reviewer's valuable comments. We have revised the manuscript accordingly. We invite further consideration of our manuscript for publication in *Nature Communications*.

Additionally, I have the following minor comments on the work:

Minor Comment 1. I did not realize based on the title + abstract that the authors aim for direct measurements vs. secondary measurements of the differences in the films. Please consider stating this more clearly already early in the paper, since to me it is the most interesting aspect of the work.

Response: Thanks for the reviewer's comment. We have revised the abstract in the revised manuscript.

Changes in the manuscript: Understanding the function of moisture on perovskite is challenging since the random environmental moisture strongly disturbs the perovskite structure. Here, we develop various N₂-protected characterization techniques to comprehensively study the effect of moisture on the efficient cesium, methylammonium (MA), and formamidinium (FA) triple-cation perovskite

$(\text{Cs}_{0.05}\text{FA}_{0.75}\text{MA}_{0.20})\text{Pb}(\text{I}_{0.96}\text{Br}_{0.04})_3$. In contrast to the secondary measurements, the established air-exposure-free techniques allow us directly monitor the influence of moisture during perovskite crystallization. We find a controllable moisture treatment for the intermediate perovskite can promote the mass transportation of organic salts, and help them enter the buried bottom of the films. This process accelerates the quasi-solid-solid reaction between organic salts and PbI_2 , enables a spatially homogeneous intermediate phase, and translates to high-quality perovskites with much-suppressed defects. Consequently, we obtain a champion device efficiency of approaching 24% with negligible hysteresis. The devices exhibit an average T_{80} -lifetime of 852 h (maximum 1210 h) working at the maximum power point.

Minor Comment 2. Some of the measurements seem to have been performed practically in-situ during the film preparation and solidification, some have required transporting the samples in the specifically prepared protective containers. Also solar cell preparation requires additional stages of work which takes time. Currently it is not clear how similar/different are the storage times under N_2 between the different measurements performed to films at the same preparation stage. Please comment on this.

Response: Thanks for the reviewer's comment. For the solar cell preparation, the storage time of the samples under N_2 is about 5 min. After the second-step spin-coating, all the samples are stored in N_2 and then transferred for moisture treatment. The UV-Vis and PL measurements have been performed without the additional transfer process. The retention time of samples in N_2 is intentionally kept similar to that during the solar cell preparation. For the XRD, SEM, and TGA measurements, the sample is transported via N_2 -protected containers, and the time required before testing is less than 10 min. The evolution of UV-vis spectra as a function of time for the pristine and the target film (Figure 2f and 2g) shows that the last several spectra are very similar to each other, implying the minor time mismatch will most probably not lead to a major difference. Particularly, for the XPS measurements, the storage time is as long as 12 h due to the pumping for high vacuum. The pristine sample shows inhomogeneous compositional

distribution after a long time staying in N₂, further indicating the barrier of layer-to-layer species diffusion. For the target sample with moisture treatment, it is believed that the cation redistribution mostly occurs in the first 5 min of moisture treatment other than during staying in the vacuum. Therefore, we believe that the difference in the storage times under N₂ is likely not to lead to a major influence.

Minor Comment 3. Supplementary Figure 9: Is the sample currently labeled as "Annealed" actually "Over humidified and annealed"?

Response: Thanks for the reviewer's comment. Yes, the sample currently labeled as "Annealed" is "Over humidified and annealed" and now it is corrected in the revised SI.

REVIEWERS' COMMENTS

Reviewer #1 (Remarks to the Author):

The revised manuscript has well responded to all the comments given by the reviewers and supplemented essential experimental data, thus it is can be accepted for publication without further revision

Reviewer #2 (Remarks to the Author):

Thanks to the authors for addressing my comments. I would recommend publishing this manuscript.

Reviewer #3 (Remarks to the Author):

After the major revisions (both further analysis and experiments, fulfilling all the requests I had in the previous review stage), I am in support of publishing this manuscript without further revisions.